# SeamCrafter: Enhancing Mesh Seam Generation for Artist UV Unwrapping via Reinforcement Learning

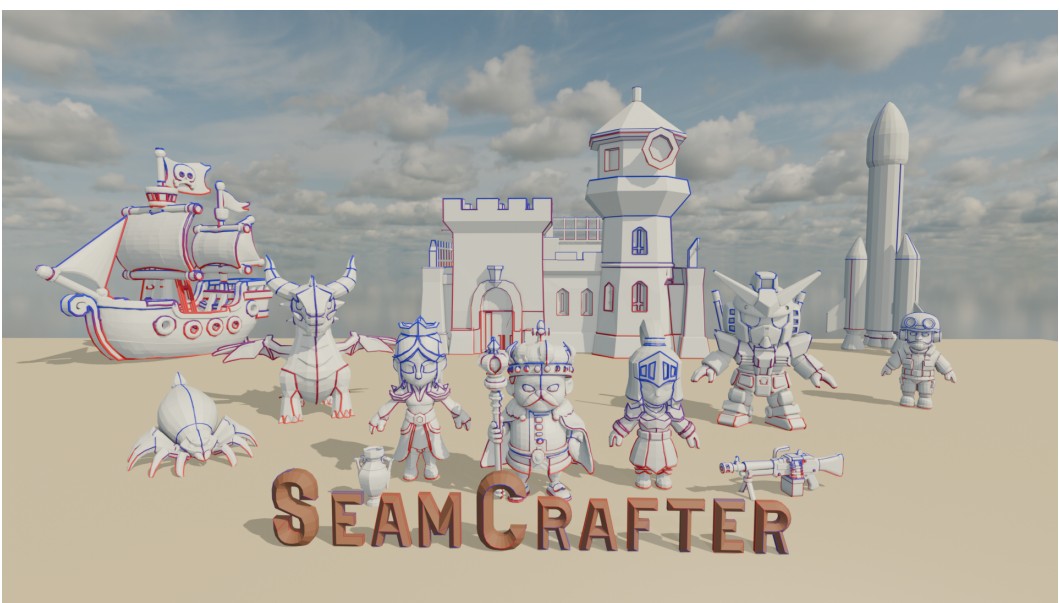

Figure 1: SeamCrafter generates artist-quality seams on diverse meshes, from characters to hard-surface models, balancing topology alignment with geometric coverage.

## Abstract

Mesh seams play a pivotal role in partitioning 3D surfaces for UV parametrization and texture mapping. Poorly placed seams often result in severe UV distortion or excessive fragmentation, thereby hindering texture synthesis and disrupting artist workflows. Existing methods frequently trade one failure mode for another—producing either high distortion or many scattered islands. To address this, we introduce SeamCrafter, an autoregressive GPT-style seam generator conditioned on point cloud inputs. SeamCrafter employs a dual-branch point-cloud encoder that disentangles and captures complementary topological and geometric cues during pretraining. To further enhance seam quality, we fine-tune the model using Direct Preference Optimization (DPO) on a preference dataset derived from a novel seam-evaluation framework. This framework assesses seams primarily by UV distortion and fragmentation, and provides pairwise preference labels to guide optimization. Extensive experiments demonstrate that SeamCrafter produces seams with substantially lower distortion and fragmentation than prior approaches, while preserving topological consistency and visual fidelity.

## 1 Introduction

Mesh seams, the edges on a 3D surface that define where the surface is cut and unfolded into a 2D UV domain, play a central role in UV parameterization and texture mapping. Well-chosen seams

enable faithful texture alignment with minimal distortion, while poorly placed seams cause stretching, compression, and visible discontinuities. Because seams partition a surface into UV islands, an ideal configuration both reduces fragmentation and produces compact, semantically coherent islands, thereby supporting efficient texture synthesis and artist-friendly editing workflows.

A rich body of prior work has explored seam generation through geometric heuristics and global optimization. Classical approaches such as region-growing (Sorkine et al., 2002; Yamauchi et al., 2005; Zhou et al., 2004) and variational formulations (Sharp & Crane, 2018) aim to balance distortion against cut complexity, but they frequently produce fragmented UV atlases and are sensitive to initialization and parameter choices. Alternative strategies—geometry images (Gu et al., 2002) and field-based methods (Lucquin et al., 2017)—regularize surfaces into structured domains but often incur substantial computational cost and limited robustness across diverse shapes. Motivated by the success of learning-based methods in related tasks, recent work has begun to explore neural seam prediction (Groueix et al., 2018; Srinivasan et al., 2024; Zhang et al., 2024; Li et al., 2025). In particular, autoregressive formulations such as SeamGPT (Li et al., 2025) represent a promising data-driven direction. However, SeamGPT tends to over-rely on topological cues while lacking sufficient geometric awareness, and its outputs often misalign with human aesthetic and practical preferences. As a result, it frequently generates redundant or fragmented cuts that degrade UV-friendliness for downstream tasks such as parameterization and texture mapping.

To address these shortcomings, we propose SeamCrafter, a novel autoregressive seam generator that (a) jointly leverages both topological and geometric cues of the input mesh, and (b) aligns predictions with human preferences, *i.e.*, seams that are UV-friendly for downstream texture workflows. To better encode input mesh, we introduce a dual-branch encoder that disentangles geometry and topology: we uniformly sample points on the mesh surface to capture local geometric detail and sample points from the vertex–edge skeleton to capture topological structure. And each stream is processed by a VecSet–based point-cloud encoder (Zhang et al., 2023) to produce complementary representations. This enriched representation substantially improves robustness and generalization across both artist-created and AI-generated meshes. Our training is divided into two stages. We first conduct supervised pretraining on large-scale seam data to learn generalizable mappings from meshes to seam layouts. Yet, supervised learning alone cannot capture the nuanced trade-offs between distortion and fragmentation, which are subjective and difficult to encode in explicit loss functions. To address this, we introduce a second stage based on Direct Preference Optimization (DPO) (Rafailov et al., 2023), leveraging a curated dataset of pairwise seam comparisons derived from our evaluation framework. By aligning the model with preference signals that reflect desirable properties—low distortion, reduced fragmentation, and compact UV islands—DPO bridges the gap between quantitative metrics and human judgments, guiding SeamCrafter toward seams that are both UV-faithful and practical for downstream tasks.

In summary, our contributions are as follows:

- We propose a dual-branch encoder that disentangles geometric and topological aspects of the input mesh, enabling richer shape understanding and improved generalization.
- We introduce a seam evaluation framework that provides pairwise preference signals, and apply Direct Preference Optimization to align the model with human judgments of seam quality.
- We demonstrate through extensive experiments that SeamCrafter produces seams with lower UV distortion and markedly reduced fragmentation compared to prior approaches.

## 2 RELATED WORK

### 2.1 TRADITIONAL APPROACHES TO SEAM GENERATION

Early research on seam construction was primarily motivated by surface parameterization and texture mapping. Classical approaches relied on geometric heuristics to balance distortion and cut complexity. For instance, region-growing strategies (Sorkine et al., 2002; Yamauchi et al., 2005; Zhou et al., 2004) iteratively expand local patches until distortion thresholds are reached, forming the basis of widely used tools such as XAtlas (Young, 2022) and Blender's Smart UV unwrapper. While effective in practice, these approaches often produce fragmented atlases and provide limited semantic control. To address these limitations, later works formulated seam placement as a global optimization problem. Variational formulations (Sharp & Crane, 2018), for example, aim to mini-

mize conformal distortion under user-specified or automatically generated seam layouts. Similarly, OptCuts (Li et al., 2018) treats cutting and mapping as a coupled task, optimizing cut length and distortion jointly. Beyond explicit seam optimization, alternative representations have been explored to regularize surface geometry into more structured forms. Geometry images (Gu et al., 2002) and their multi-chart extensions (Sander et al., 2003; Carr et al., 2006) map meshes to grid-like domains, facilitating storage and downstream processing. Building on a different principle, field-based methods such as SeamCut (Lucquin et al., 2017) automate segmentation without relying on mesh topology. Despite their mathematical elegance, these approaches often need complex optimization and are sensitive to initialization and parameters, limiting their robustness in practice.

## 2.2 Learning-based Approaches to Seam Generation

With the advent of deep learning, data-driven methods have been proposed to predict or refine UV decompositions. Rather than solving each instance independently, neural models learn reusable priors that capture how surfaces can be partitioned and unfolded. Early explorations include neural atlas approaches such as AtlasNet (Groueix et al., 2018), where local surface patches are parameterized using implicit functions or neural fields. Extending this idea, Nuvo (Srinivasan et al., 2024) formulates parameterization as optimizing a set of neural fields under structural constraints, enabling more flexible surface representations. Subsequent work has sought greater modularity and consistency. For example, FAM (Zhang et al., 2024) decomposes the task into three dedicated subnetworks—cutting, deformation, and unwrapping—organized within a cycle-mapping framework to enforce mutual consistency. These methods demonstrate that neural networks can capture structural regularities across shapes. However, most still require extensive per-scene fine-tuning and lack explicit mechanisms for preserving semantic coherence in the predicted seams. More recently, SeamGPT (Li et al., 2025) introduces an autoregressive formulation that is conceptually close to our setting. Nevertheless, it shows weak geometric awareness and heavy reliance on topology, often producing fragmented seams, redundant cuts, and limited generalization across diverse meshes.

## 2.3 UV Unwrapping Approaches

UV unwrapping aims to map 3D mesh surfaces onto 2D domains with minimal distortion, and has traditionally been formulated as a numerical optimization problem. Classical methods, such as LSCM (Lévy et al., 2023) and ABF++ (Sheffer et al., 2005), assume pre-defined seams and optimize angular distortion, while bijective optimization techniques like SLIM (Rabinovich et al., 2017) and SCAF (Jiang et al., 2017) explicitly enforce overlap constraints. Joint approaches, such as region-growing strategies (Sorkine et al., 2002) or OptCuts (Li et al., 2018), attempt to optimize seams and parameterization simultaneously, but often lead to fragmented or semantically incoherent UV maps. More recently, learning-based methods such as Nuvo (Srinivasan et al., 2024) and FAM (Zhang et al., 2024) decompose unwrapping into subtasks and leverage neural priors.

# 3 Method

Figure 2 presents an overview of our proposed framework, which consists of two main stages: pretraining and post-training. In the pretraining stage, we employ supervised learning on large-scale training data to establish a strong initialization of the model. Subsequently, in the post-training stage, we adopt the Direct Preference Optimization (Rafailov et al., 2023) method to further fine-tune the model. The fine-tuning process leverages a preference dataset constructed through our proposed seam evaluation system, enabling the model to better align with human judgments of seam quality.

## 3.1 Pre-training of SeamCrafter

### 3.1.1 Mesh Seam Tokenization

Following SeamGPT (Li et al., 2025) and wireframe generation (Ma et al., 2024), we represent mesh seams as a sequence of segments $S_{\mathcal{M}} = \{s_i\}_{i=1}^{N}$. Each segment $s_i = [p_{i,1}, p_{i,2}] \in \mathbb{R}^{2 \times 3}$ is defined by the 3D coordinates of its two endpoints, forming a hierarchical representation that consists of seam segments, endpoints, and endpoint coordinates. To enable auto-regressive modeling, all vertex coordinates are quantized into 1024 discrete bins and ordered according to a $yzx$ scheme, where

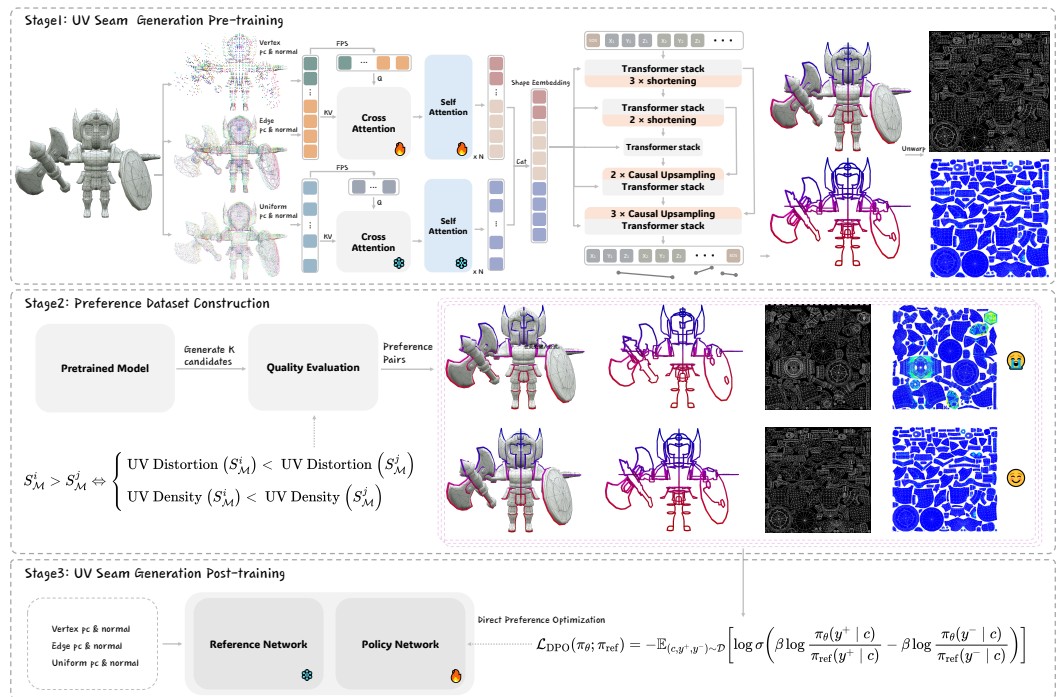

Figure 2: An overview of SeamCrafter. The pipeline consists of three stages: (1) supervised pretraining with point clouds and ground-truth seam sequences; (2) candidate seam generation and quality assessment via our seam evaluation system to construct a preference dataset; and (3) post-training with Direct Preference Optimization (DPO) to refine the model.

the $y$-axis corresponds to the vertical direction. The two endpoints of each segment are further arranged in ascending order using a lexicographic rule. Finally, the entire set of seam segments is ordered according to the first endpoint of each segment. During UV unwrapping, the predicted seam segments are projected back onto the mesh. Specifically, the endpoints of each segment are projected onto the nearest mesh vertices, and a topological edge path connecting these vertices is identified and marked as seam edges. The mesh surface is then split and unfolded along the marked seams to complete the unwrapping process.

### 3.1.2 SHAPE GEOMETRIC AND TOPOLOGICAL ENCODER

In this work, we address UV seam generation from 3D meshes. To encode the input mesh, we adopt a VecSet-based point-cloud encoder (Zhang et al., 2023), as VecSet-style encoders have shown strong performance in 3D shape representation (Hao et al., 2024; Zhao et al., 2025; Weng et al., 2025). Given a point cloud sampled from the mesh, the encoder selects farthest-point sampled (FPS) anchors and applies a point-query (cross-attention) mechanism with the full context, producing latent tokens as conditional input to the seam generator. We next revisit point-cloud sampling strategies to obtain more effective conditioning for UV seam generation.

SeamGPT (Li et al., 2025) samples points on vertices and edges instead of uniformly over the surface, making seam predictions overly dependent on mesh topology and sensitive to small topological changes. Although this captures topological cues, it overweights graph structure and under-covers surface geometry, which limits generalization to complex or highly stylized meshes. To address this limitation, we propose a dual-branch design for shape-condition encoding. Let $\mathcal{M}$ denote a mesh, we built two point sets: $\mathcal{P}_t^{\mathcal{M}} \in \mathbb{R}^{N_t \times 3}$ from mesh vertices and edges to capture topological structure, and $\mathcal{P}_g^{\mathcal{M}} \in \mathbb{R}^{N_g \times 3}$ from uniform sampling over the mesh surface to capture geometric coverage. Here, $N_t$ and $N_g$ denote the respective numbers of sampled points. We encode both point sets with two VecSet encoders of identical architectures but separate parameters:

$$\mathcal{E}_t : \mathbb{R}^{N_t \times 3} \rightarrow \mathbb{R}^{\ell \times d}, \quad \mathcal{E}_g : \mathbb{R}^{N_g \times 3} \rightarrow \mathbb{R}^{\ell \times d}, \tag{1}$$

and obtain the final latent shape embedding by concatenating the two encoded outputs:

$$e_{\mathcal{M}} = \left[ \mathcal{E}_t(\mathcal{P}_t^{\mathcal{M}}) \, \| \, \mathcal{E}_g(\mathcal{P}_g^{\mathcal{M}}) \right] \in \mathbb{R}^{(2\ell) \times d}, \qquad (2)$$

where $l$ denotes the number of tokens per branch and $d$ is the token dimensionality. This simple fusion provides a more robust conditioning for artistic-style meshes, enabling the model to generate seam layouts that are both concise and structurally rational, while also improving generalization to denser meshes or those with suboptimal wiring.

### 3.1.3 HOURGLASS MESH SEAM DECODER

Similar to (Li et al., 2025; Hao et al., 2024), we employ an autoregressive hourglass transformer decoder (Nawrot et al., 2021) which enables multiple levels sequence abstraction to generate the endpoint coordinate codes of seam segments directly. The 3D shape condition embeddings are injected into the first layer of the transformer stack at each level of the hourglass architecture via cross-attention. The input coordinate code sequences are first downsampled by a factor of three at the coordinate level and further downsampled by a factor of two at the segment-endpoint level, followed by corresponding upsampling operations with factors of two and three, respectively. Both downsampling and upsampling are implemented through a causality-preserving mechanism.

### 3.2 POST-TRAINING OF SEAMCRAFTER

Although our pre-trained model can generate high-quality mesh seams for UV unwrapping, it occasionally produces over-segmented or highly distorted seams, which can hinder downstream tasks such as texture mapping and editing. To further improve performance, we design a comprehensive evaluation pipeline to curate a preference dataset and employ Direct Preference Optimization (DPO) (Rafailov et al., 2023) to align seam generation with human preferences, emphasizing both high UV fidelity and low fragmentation.

### 3.2.1 MESH SEAMS EVALUATION SYSTEM

Constructing high-quality preference datasets is essential for effective direct preference learning. However, manually selecting preference pairs is prohibitively time-consuming. To address this, we introduce a set of automatic metrics for evaluating the quality of generated samples. Inspired by optimization-based UV unwrapping methods, we primarily assess the quality of generated mesh seams in UV space. Specifically, our seam evaluation system comprises two components, which encourage preferences for mesh seams that yield UV mappings with high fidelity and low fragmentation density.

**UV Mapping Fidelity.** Fidelity refers to the accuracy of the UV mapping process in preserving the details and proportional relationships of the original 3D mesh. A high level of fidelity indicates that the projection of textures onto the model's surface does not introduce noticeable overlaps or distortions. We quantify fidelity by measuring the amount of distortion, which is commonly used as an energy term in optimization-based methods. The distortion is computed over the UV mapping using the symmetric Dirichlet energy normalized by the surface area, and is formulated as follows:

$$Distortion(S_{\mathcal{M}}) = \frac{1}{\sum_{t \in \mathcal{F}} |A_t|} \sum_{t \in \mathcal{F}} |A_t| \left\| \sigma_{t,1}^2 - \sigma_{t,2}^2 \right\|_1, \qquad (3)$$

where $\mathcal{F}$ is the set of all triangles, $|A_t|$ is the area of triangle $t$ on the input surface, and $\sigma_{t,i}$ is the $i$-th singular value of the deformation gradient of triangle $t$.

**UV Mapping Fragmentation Density.** A high fragmentation density indicates that the UV space or model surface is subdivided into numerous small islands. Such fragmentation complicates texture painting, as artists must constantly switch between scattered islands, disrupting continuity and efficiency. To quantify this property, we simply compute the number of UV islands as a measure of fragmentation, encouraging fewer islands under the same level of distortion. We denote $Density(S_{\mathcal{M}})$ as the number of UV islands in $S_{\mathcal{M}}$.

### 3.2.2 PREFERENCE DATASET CONSTRUCTION

We employ our proposed mesh seam evaluation system to construct a dataset of preference pairs. Given an input point cloud $\mathcal{P}_{\mathcal{M}}$ sampled from a mesh $\mathcal{M}$, the pre-trained model first generates

five candidate mesh seams $\left\{S_{\mathcal{M}}^i\right\}_{i=1}^5$. These candidates are then evaluated by our seam evaluation system. A candidate $S_{\mathcal{M}}^i$ is considered a positive sample and $S_{\mathcal{M}}^j$ a negative sample if and only if $S_{\mathcal{M}}^i$ strictly outperforms $S_{\mathcal{M}}^j$ across all evaluation metrics, *i.e.*,

$$S_{\mathcal{M}}^i \succ S_{\mathcal{M}}^j \iff \begin{cases} Distortion(S_{\mathcal{M}}^i) < Distortion(S_{\mathcal{M}}^j) \\ Density(S_{\mathcal{M}}^i) < Density(S_{\mathcal{M}}^j) \end{cases}, \tag{4}$$

Each such positive–negative pair constitutes a preference pair, where we denote $S_{\mathcal{M}}^+$ as the positive sample and $S_{\mathcal{M}}^-$ as the negative sample. Figure 3 illustrates examples of the collected preference pairs. In total, we construct a mesh seam preference dataset consisting of 4,000 preference pairs, which is subsequently used to support the post-training of DPO.

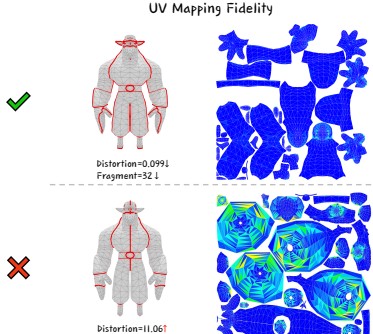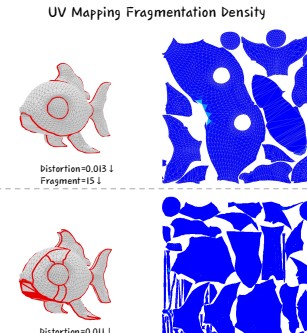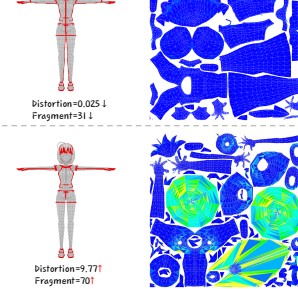

Figure 3: Some examples of the collected preference pairs annotate based on the mesh seams evaluation system. (In the UV map, regions with brighter yellow colors correspond to higher distortion.)

### 3.2.3 DIRECT PREFERENCE OPTIMIZATION

Direct Preference Optimization (DPO) is a recently proposed training paradigm for aligning models with human preferences. Unlike traditional approaches that rely on separately trained reward models or reinforcement learning, DPO provides a closed-form objective that directly optimizes the policy using preference data. Specifically, it fine-tunes the pre-trained model by minimizing the following loss:

$$\mathcal{L}\left(\pi_\theta; \pi_{\text{ref}}\right) = -\mathbb{E}_{\left(\mathcal{P}_{\mathcal{M}}, S_{\mathcal{M}}^+, S_{\mathcal{M}}^-\right) \sim \mathcal{D}} \left[\log \sigma \left(\beta \log \frac{\pi_\theta\left(S_{\mathcal{M}}^+ \mid \mathcal{P}_{\mathcal{M}}\right)}{\pi_{\text{ref}}\left(S_{\mathcal{M}}^+ \mid \mathcal{P}_{\mathcal{M}}\right)} - \beta \log \frac{\pi_\theta\left(S_{\mathcal{M}}^- \mid \mathcal{P}_{\mathcal{M}}\right)}{\pi_{\text{ref}}\left(S_{\mathcal{M}}^- \mid \mathcal{P}_{\mathcal{M}}\right)}\right)\right] \tag{5}$$

where $\beta$ is a scaling factor that balances the contributions of preferred and dispreferred samples, $\sigma(\cdot)$ denotes the sigmoid function, and $\mathcal{D}$ represents the constructed preference dataset. Here, $\pi_{\text{ref}}$ denotes the token-level probability distribution given by the reference model, which is identical to the pre-trained model and kept frozen, while $\pi_\theta$ denotes the token-level probability distribution produced by the trainable policy model, which is initialized from the same pre-trained model. Using this objective, we post-train the pre-trained model on our constructed preference pair dataset, thereby encouraging the model to favor candidates with lower distortion, lower fragmentation density, and higher semantic consistency.

## 4 EXPERIMENTS

### 4.1 PRE-TRAINING DATA CURATION

Building on the data pipeline introduced in SeamGPT (Li et al., 2025), we constructed our training corpus from several large-scale open-source 3D datasets, including Objaverse (Deitke et al., 2022), Objaverse-XL (Deitke et al., 2023), and 3D-FUTURE (Fu et al., 2021). Our curation primarily targeted meshes with valid UV coordinates, since these provide artist-defined seam annotations that are critical for our task. To ensure data quality, we extended the original SeamGPT filtering strategy with additional repair and refinement procedures. Specifically, we discarded meshes with

poor topology and eliminated those with problematic UV unwrappings, such as overlapping islands, arbitrary seams lacking semantic relevance, or excessive overcutting seams. Beyond filtering, we introduced automated consistency checks and corrective operations to repair partially corrupted UV layouts, thereby preserving structural coherence between seams and the underlying mesh geometry. After this multi-stage pipeline, we obtained a large-scale training set of approximately 700K meshes. To further enhance generalization, we applied controlled geometric transformations, where meshes were rotated around the $y$-axis in $10°$ increments, uniformly scaled within $[0.9, 1.1]$, and randomly translated within $[-0.1, 0.1]$.

## 4.2 IMPLEMENTATION DETAILS

Our Hourglass Transformer model consists of 24 layers, with a repeating pattern of three self-attention layers followed by one cross-attention layer conditioned on the point cloud features $e_{\mathcal{M}}$. For each mesh input, we sample $N_g = N_t = 30,720$ points for the geometry and topology branches, respectively. Each branch produces $l = 3,072$ tokens of dimensionality $d = 1,024$. During training, the geometry point cloud encoder $\mathcal{E}_g$ is kept fixed, initialized from pre-trained weights learned on a large mesh dataset, while the topology point cloud encoder $\mathcal{E}_t$ is trained jointly with the rest of the model. Pretraining is performed on 64 Nvidia H20 GPUs (98 GB memory per GPU) for 200K steps with a batch size of 128. Post-training is conducted on 16 H20 GPUs for 2,500 steps using a fixed learning rate of $1 \times 10^{-6}$. At inference, the model can run on most consumer-grade GPUs with a memory footprint not exceeding 16 GB, demonstrating both scalability and practical deployability.

## 4.3 SURFACE SEAM PROJECTION

In our framework, the model predicts the 3D coordinates of the two endpoints for each candidate cut line. To enable reliable UV unwrapping, these predicted lines must be consistently mapped onto the mesh surface, with their corresponding edges designated as seams. To achieve this, we first construct a topological graph representation of the input mesh, where vertices correspond to mesh points and edges capture their connectivity. For each predicted cut line, we identify the two surface points that are closest to its endpoints via a nearest-neighbor search. Using these surface correspondences as anchors, we compute the shortest geodesic path between the two points over the constructed graph. All mesh edges along this path are then labeled as seams, thereby ensuring that the predicted cuts are faithfully transferred to the mesh topology for subsequent UV parameterization.

## 4.4 BENCHMARK AND METRIC

We conducted a comprehensive comparison between our proposed method and several state-of-the-art approaches on both publicly available benchmarks and our newly constructed AIGC-100 test set. The public benchmarks include Toys4K (Stojanov et al., 2021), Shapenet (Chang et al., 2015), and FAM (Zhang et al., 2024), whereas AIGC-100 is composed of 100 AI-generated 3D objects. To ensure a rigorous evaluation, we adopted three widely used metrics: distortion, fragmentation, and runtime. In particular, distortion is defined as the average conformal energy across all triangular faces, while fragmentation is characterized by the number of UV islands present in the UV map.

## 4.5 COMPARISON RESULTS

Figure 4 highlights three advantages of SeamCrafter over XAtlas (Young, 2022), FAM (Zhang et al., 2024), Nuvo (Srinivasan et al., 2024), and SeamGPT (Li et al., 2025): (1) compact yet complete seams that avoid over-segmentation; (2) sufficient coverage that prevents large uncut regions and thus mitigates flattening-induced distortion; and (3) semantic consistency across islands. These gains are a direct consequence of our dual-branch point-cloud encoder, which separately encodes topological skeletons and uniformly sampled surface geometry to provide complementary, globally consistent cues, and our preference-aligned post-training (DPO) that optimizes the trade-off between distortion and fragmentation using pairwise preferences from our evaluation framework. In contrast, heuristic region-growing methods (e.g., XAtlas) depend on local thresholds, leading to over-fragmentation or missed semantically meaningful cuts; FAM and Nuvo optimize UVs directly on vertices, remain agnostic to mesh connectivity and wiring, and therefore lack mesh-structure constraints, which can lead to severe distortion, especially on complex or irregular geometry; and

Figure 4: Qualitative UV flattening results. Compared with existing methods, our approach produces seams with lower distortion and fewer fragmented UV islands, yielding more coherent and visually consistent layouts.

topology-heavy AR predictors (SeamGPT) under-utilize geometry, often producing redundant cuts and unstable coverage. As a result, SeamCrafter yields more coherent, well-structured, and editing-friendly UV maps suitable for professional workflows.

| | Toys4k | | | Shapenet | | |
|---|---|---|---|---|---|---|
| | Distortion ↓ | Fragments ↓ | Runtime(s) ↓ | Distortion ↓ | Fragments ↓ | Runtime(s) ↓ |
| XAtlas | 1.86 | 44.71 | **19.28** | 35.82 | 800.43 | **18.58** |
| Nuvo | 11.43 | / | 2569.06 | 174.91 | / | 2352.48 |
| Fam | 8.05 | / | 3090.06 | 599.02 | / | 2609.94 |
| SeamGPT | 1.85 | 7.86 | 446.11 | 37.51 | 183.13 | 487.22 |
| Ours w/o DPO | 1.41 | 11.57 | 51.43 | 34.19 | 100.78 | 17.46 |
| Ours w DPO | **1.39** | **7.28** | 50.67 | **32.60** | **91.28** | 22.07 |

| | FAM-benchmark | | | AIGC-100 | | |
|---|---|---|---|---|---|---|
| | Distortion ↓ | Fragments ↓ | Runtime(s) ↓ | Distortion ↓ | Fragments ↓ | Runtime(s) ↓ |
| XAtlas | 10.51 | 74.67 | 80.36 | 14.75 | 135.42 | **2.56** |
| Nuvo | 20.37 | / | 2925.75 | 70.84 | / | 2154.23 |
| Fam | 12.33 | / | 5656.25 | 42.23 | / | 3544.78 |
| SeamGPT | 11.96 | 30.33 | 107.13 | 14.43 | 68.77 | 29.39 |
| Ours w/o DPO | 11.85 | 27.33 | 57.13 | 13.65 | 45.12 | 8.73 |
| Ours w DPO | **10.07** | **10.05** | **59.66** | **10.63** | **33.72** | 13.55 |

Table 1: Quantitative evaluation on Toys4k, ShapeNet, FAM-benchmark, and AIGC-100. Our method achieves superior performance in terms of UV distortion, number of fragments, and runtime compared to prior approaches.

The quantitative results are shown in Table 1, where our algorithm outperforms other methods in all four benchmarks. Specifically, our method without DPO already achieves low distortion and fragmentation across all benchmarks—for instance, 1.41 distortion and 11.57 fragments on Toys4k—while maintaining competitive runtime. Adding DPO further improves performance, re-

ducing distortion and fragments even more (1.39 and 7.28 on Toys4k) without significantly affecting runtime. Compared to XAtlas and SeamGPT, which either produce more fragments or run slower, our approach consistently provides a superior balance between quality and efficiency, demonstrating its effectiveness and robustness across diverse datasets.

## 4.6 ABLATION STUDY

To rigorously assess the adequacy and effectiveness of the proposed Mesh Seams Evaluation System, we conducted both qualitative and quantitative evaluations. As shown in Figure 5 and Table 2, when the model is used without DPO training, it sometimes fails to generate sufficient seams in certain regions, which leads to excessive UV map distortion. In other cases, it produces overly dense seams, resulting in severe fragmentation of UV islands.

When DPO training pairs are built solely based on the Distortion metric, the finetuned model suppresses UV map distortion but introduces an excessive number of seams, causing highly fragmented UV islands. Conversely, when optimizing exclusively with respect to the Fidelity metric, fragmentation is mitigated, yet the reduced number of seams results in significant UV map distortion.

By jointly considering both metrics during DPO training, our method achieves a principled trade-off between distortion and fragmentation. This optimization produces UV maps with low distortion and well-preserved island integrity, while simultaneously enhancing the model's robustness and aligning it with human judgments of seam quality.

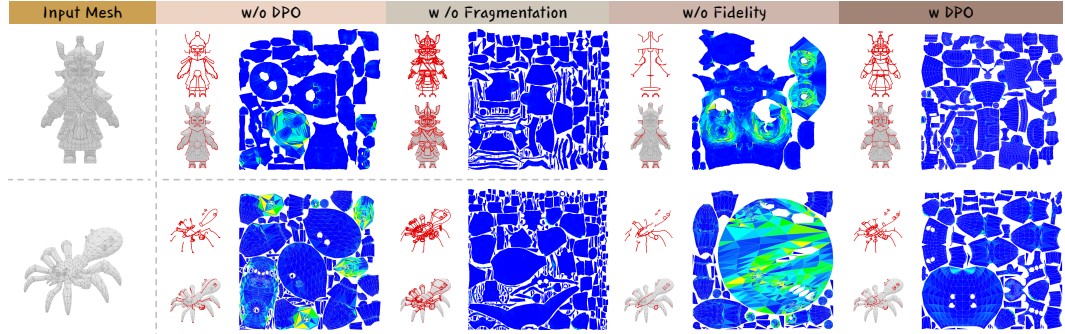

Figure 5: Qualitative results from the ablation study on DPO training. Joint optimization of Distortion and Fidelity produces seams that balance low UV distortion with coherent, compact UV islands, unlike models trained without DPO or with single-metric optimization.

|  | w/o DPO | w/o Fidelity | w/o Fragmentation | w DPO |
|---|---|---|---|---|
| Distortion ↓ | 13.65 | 72.80 | 2.18 | 10.63 |
| Fragments ↓ | 45.12 | 22.72 | 127.20 | 33.72 |

Table 2: Quantitative ablation results on the effect of DPO metrics. Optimizing only Distortion or Fidelity leads to extreme UV distortion or fragmentation, while joint DPO achieves a principled trade-off with both low distortion and reduced fragmentation.

## 5 CONCLUSION

We presented SeamCrafter, an autoregressive framework for UV-friendly mesh seam generation that combines a dual-branch encoder to disentangle geometry and topology with Direct Preference Optimization (DPO) guided by a curated seam evaluation dataset. This design enables the model to capture both local detail and global shape semantics while aligning predictions with human judgments of seam quality. Experiments show that SeamCrafter achieves lower distortion, reduced fragmentation, and stronger semantic coherence compared to prior approaches. In future work, we plan to extend preference-guided optimization to downstream tasks such as texture editing and animation.

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
