# OpenReview forum: "SeamCrafter: Enhancing Mesh Seam Generation for Artist UV Unwrapping via Reinforcement Learning"
_ICLR.cc/2026/Conference — ICLR 2026 Conference Withdrawn Submission_

### Official Review · Reviewer_LECe · 2025-10-31

**Soundness:** 3
**Presentation:** 3
**Contribution:** 3
**Rating:** 4
**Confidence:** 3

**Summary:**

The paper introduces SeamCrafter, an autoregressive GPT-style model designed to generate high-quality mesh seams for UV unwrapping and texture mapping. The goal is to produce "artist-quality" seams that balance low UV distortion with reduced fragmentation, which are critical for efficient texture synthesis and artist workflows.

**Strengths:**

The core idea of using DPO to fine-tune a mesh generation model based on a custom, objective preference dataset is an application for UV unwrapping, which seems to be novel.

The explicit design of a dual-branch encoder that disentangles geometric (uniform surface sampling) and topological (vertex-edge skeleton sampling) cues for the input mesh is interesting.

**Weaknesses:**

The preference pair definition $S_{\mathcal{M}}^{i}>S_{\mathcal{M}}^{j}$ is an AND condition: $S_{\mathcal{M}}^{i}$ must strictly outperform $S_{\mathcal{M}}^{j}$ in both Distortion AND Density. This extremely strict rule limits the ability to learn the subtle, human-preferred trade-offs (e.g., a slight increase in fragmentation for a substantial decrease in distortion).

The experiment does not compare with uvatlas, which is a major oversight.

The results do not appear to be advantageous for mainstream tasks such as texturing and auto-retopology.

**Questions:**

Please see weaknesses, and provide more information.

---

### Official Review · Reviewer_WzjF · 2025-11-01

**Soundness:** 3
**Presentation:** 3
**Contribution:** 2
**Rating:** 4
**Confidence:** 4

**Summary:**

This paper proposes an auto-regressive, GPT like algorithm for placing seams towards better UV mapping. The idea is to reduce distortions and fragmentation by a two stage pipeline which is later aligned to human preferences by a preference optimization. Results demonstrate quality advantages over the existing methods.

**Strengths:**

- The approach is modern, makes sense and I believe the technical results are correct.
- The presentation is clear, even though improvements are required.
- The experimentation on several datasets seem to justify the contributions.

**Weaknesses:**

- This paper largely builds upon SeamGPT, which seems to be an unpublished work. Not sure how much this is a problem but it makes it harder to gauge correctness of certain parts. Moreover, it is not very clear what the exact contributions of this paper are, especially in relation to SeamGPT.
- The paper tries to make the seams better. However, the distortion problem still exists even with the best choices of seams. Some methods instead opted for methods that learn to represent textures directly on 3D surfaces / points bypassing the UV altogether. I think this paper should discuss some of these related works:
* Foti, S., Zafeiriou, S., & Birdal, T. Uv-free texture generation with denoising and geodesic heat diffusion. NeurIPS 2024.
* Wang, X., Cheng, Z., ... & Li, H. DoubleDiffusion: Combining Heat Diffusion with Denoising Diffusion for Texture Generation on 3D Meshes. arXiv preprint arXiv:2501.03397. 2025.

- I would like to see a discussion and comparison as to why the fidelity and density metrics are not used as regularizers and optimized for as done in previous works. What is it about DPO that makes it special for this case?
- I would have expected some (maybe small) dataset that is artist curated (maybe through some interaction) that acts as a guide to align the output with artist preferences. Otherwise, we rely completely on the density/fidelity metrics to capture the full preferences of the humans/artists, which I believe is not realistic.
- The paper stresses too much the topology aspect but then uses a point cloud encoder to encode some skeleton, completely ignoring the mesh structure, which also provides topological information. Is there a rationale behind this choice? Can we see comparisons?
- Seams seem to be snapped to mesh vertices / edges. When elongated and potentially large triangles are present, this might lead to failures. Thus, I would ask for a comparison with different mesh structures where triangle distributions are non uniform.
- Paper really has very few related works and seems to ignore a very large literature on UV-mapping and texturing. I would strongly suggest the authors to conduct a thorough literature survey. These topics are not new.

Minor issues:
H20 has 96GB memory, not 98, to the best of my knowledge.

**Questions:**

Please see weaknesses, where the questions are also asked. I am particularly unclear about the differences w.r.t. SeamGPT (except the point sampling) and what the paper truly contributes other than some implementation / design choices.

---

### Official Review · Reviewer_T99u · 2025-11-01

**Soundness:** 3
**Presentation:** 3
**Contribution:** 2
**Rating:** 6
**Confidence:** 3

**Summary:**

This paper proposes SeamCrafter, a method for generating high quality parameterization seams on meshes. The method consists of two stages. First the model is pretrained using large scale data from multiple open source 3D datasets. In the second stage, the model is fine-tuned through direct preference optimization (DPO) to better align to human preferences. In order to train the model with DPO, the authors automatically create a custom dataset of preference pairs that mirrors human preferences by using both a distortion and density metric. The method is evaluated qualitatively through comparison figures and quantitatively with distortion and fragment metrics, outperforming baseline approaches on both. The authors also include an ablation study on the different components of their DPO to motivate the importance of each one.

**Strengths:**

- Disentangles geometry and topology using a two branched encoder.
- Qualitative results highlight clear improvements over baseline methods while quantitative metrics also show significant decreases in distortion and cuts/fragments.
- Training on a large dataset of artist-created parameterizations enables a robust and generalizable model.

**Weaknesses:**

- Since aiming to produce seams that are desired by real artists, it would be helpful to include a user study that records how real artists rate each of these methods.
- My understanding is that normally DPO is used to incorporate human feedback into the training. However, the metrics used to create the pairs (distortion and density) are entirely automated. Could one just train with an additional loss consisting of these distortion and density metrics? Adding an ablation on some approach like this could make the justification for using DPO stronger.

**Questions:**

See weaknesses.

---

### Official Review · Reviewer_2fG6 · 2025-11-01

**Soundness:** 2
**Presentation:** 3
**Contribution:** 2
**Rating:** 4
**Confidence:** 4

**Summary:**

This paper introduces SeamCrafter, a new method for generating high-quality mesh seams for 3D UV unwrapping. The approach utilizes an autoregressive, GPT-style transformer that is conditioned on a dual-branch point cloud encoder, designed to capture both topological structure and geometric surface detail. The core novelty lies in its two-stage training process: a supervised pre-training stage on a large-scale dataset, followed by a fine-tuning stage using Direct Preference Optimization (DPO). To enable DPO, the authors construct a new preference dataset by automatically evaluating seam candidates based on a framework that measures UV distortion and fragmentation density. Experiments show that SeamCrafter outperforms existing heuristic and learning-based methods in producing seams that better balance these two competing objectives.

**Strengths:**

1. The proposed dual-branch encoder, which disentangles topological and geometric information, is a thoughtful architectural improvement that demonstrably improves results over the baseline.

2. The application of Direct Preference Optimization (DPO) to this task is novel. Creating a preference dataset from automated metrics (distortion, fragmentation) is a clever way to fine-tune the model for a complex, multi-objective trade-off without requiring expensive human-in-the-loop labeling.

3. The experimental evaluation is thorough. The authors compare against a good range of baselines (heuristic, optimization-based, and learning-based) and demonstrate superior performance. The introduction of the new AIGC-100 test set to evaluate performance on AI-generated models is also a commendable contribution.

**Weaknesses:**

1. The most significant weakness is the limited perceived impact of the problem itself. UV unwrapping is a highly specialized task, and the improvements, while real, feel incremental. The work comes across as a sophisticated engineering solution to a niche problem, rather than fundamental research that would be broadly interesting or applicable to the general ICLR audience.

2. The method's robustness is not fully explored. The final step (Section 4.3) relies on projecting predicted 3D coordinates to the nearest mesh vertex and then finding a shortest geodesic path. This seems
potentially brittle. How does the method perform on meshes with complex topologies, noise, non-manifold geometry, or if the initial 3D coordinate prediction is slightly inaccurate? The reliance on a clean topological graph seems like a strong assumption.

3. The paper is missing a hyperparameter analysis. For instance, the dual-branch encoder samples $N_g=N_t=30,720$ points. How sensitive is the model to this number? Similarly, the DPO stage introduces its own set of hyperparameters (e.g., the $\beta$ in the DPO loss, the learning rate for post-training) which are not ablated.

4. The method is an extension of the autoregressive SeamGPT framework, and thus likely inherits its limitations, such as sequential generation time and potential error propagation, though this is not discussed.

**Questions:**

1. Following on the weakness mentioned above: Could you provide more details on the failure cases of the seam projection step (Section 4.3)? What happens if the nearest-neighbor search maps a predicted endpoint to the wrong vertex, or if the shortest geodesic path is not the semantically intended one?

2. The preference dataset for DPO is built on two automated metrics: distortion and density. The preference criterion (Eq. 4) is very strict, requiring a candidate to be strictly better on both metrics to be considered "preferred". Did you experiment with a more relaxed definition, for example, a weighted sum of metrics, or using a Pareto frontier? It seems this strict criterion might discard many useful comparison pairs and bias the optimization.

3. How does the model's inference time scale with mesh complexity (e.g., vertex count) and the number of generated seams, given its autoregressive nature?

---

### Note · Authors · 2025-11-13

I have read and agree with the venue's withdrawal policy on behalf of myself and my co-authors.